# Prevalence and Molecular Characterization of *Mycobacterium bovis* in Slaughtered Cattle Carcasses in Burkina Faso; West Africa

**DOI:** 10.3390/microorganisms10071378

**Published:** 2022-07-08

**Authors:** Estelle Kanyala, Yassir Adam Shuaib, Norbert Georg Schwarz, Sönke Andres, Elvira Richter, Bernard Sawadogo, Mamadou Sawadogo, Minoungou Germaine, Ouattara Lassina, Sven Poppert, Hagen Frickmann

**Affiliations:** 1Institut de Recherche en Sciences de la Santé, Bobo-Dioulasso BP 390, Burkina Faso; kanyalaestelle@gmail.com; 2College of Veterinary Medicine, Sudan University of Science and Technology, P.O. Box 204, Khartoum North 13321, Sudan; vet.aboamar@gmail.com; 3Molecular and Experimental Mycobacteriology, Research Center Borstel, 23845 Borstel, Germany; 4Independent Researcher, 67459 Böhl-Iggelheim, Germany; schwarznorbert@web.de; 5National Reference Laboratory for Mycobacteria, Research Center Borstel, 23845 Borstel, Germany; sandres@fz-borstel.de; 6Tuberculosis Laboratory, Laboratory Limbach, 69126 Heidelberg, Germany; elvira.richter@labor-limbach.de; 7West Africa Francophone, African Field Epidemiology Network (AFENET), Ouagadougou 01 BP 364, Burkina Faso; bernardsawadogo@gmail.com; 8Laboratory of Biochemistry, Health Sciences Training and Research Unit, University of Ouagadougou, Ouagadougou BP 7021, Burkina Faso; elmsawa@yahoo.fr; 9Laboratoire National d’Elevage (LNE), Ouagadougou BP 907, Burkina Faso; minoungou.germaine@gmail.com; 10Direction Générale des Services Vétérinaires (DGSv), Ouagadougou 01 BP 364, Burkina Faso; sielouattara@hotmail.com; 11Bernhard Nocht Institute for Tropical Medicine Hamburg, 20359 Hamburg, Germany; 12Department of Microbiology and Hospital Hygiene, Bundeswehr Hospital Hamburg, 20359 Hamburg, Germany; 13Institute for Medical Microbiology, Virology and Hygiene, University Medicine Rostock, 18057 Rostock, Germany

**Keywords:** Burkina Faso, cattle, *Mycobacterium bovis*, tuberculosis

## Abstract

This cross-sectional study was conducted at the slaughterhouses/slabs of Oudalan and Ouagadougou in Burkina Faso, between August and September 2013. It aimed at determining the prevalence of bovine tuberculosis (bTB) suggestive lesions in slaughtered cattle carcasses and to identify and characterize the mycobacteria isolated from these lesions. A thorough postmortem examination was conducted on carcasses of a total of 2165 randomly selected cattle. The overall prevalence of bTB suggestive lesions was 2.7% (58/2165; 95% CI 2.1–3.5%). Due to the low number of positive samples, data were descriptively presented. The lesions were either observed localized in one or a few organs or generalized (i.e., miliary bTB) in 96.6% (*n* = 57) and 3.4% (*n* = 2), respectively. The identified mycobacteria were *M. bovis* (44.4%, *n* = 20), *M. fortuitum* (8.9%, *n* = 4), *M. elephantis* (6.7%, *n* = 3), *M. brumae* (4.4%, *n* = 2), *M. avium* (2.2%, *n* = 1), *M. asiaticum* (2.2%, *n* = 1), *M. terrae* (2.2%, *n* = 1), and unknown non-tuberculous mycobacteria (NTM) (11.1%, *n* = 5). Moreover, eight mixed cultures with more than one *Mycobacterium* species growing were also observed, of which three were *M. bovis* and *M. fortuitum* and three were *M. bovis* and *M. elephantis*. In conclusion, *M. bovis* is the predominant causative agent of mycobacterial infections in the study area. Our study has identified a base to broaden the epidemiological knowledge on zoonotic transmission of mycobacteria in Burkina Faso by future studies investigating further samples from humans and animals, including wild animals employing molecular techniques.

## 1. Introduction

Bovine tuberculosis (bTB) is a chronic infectious disease of cattle [1,2]. It is zoonotic and *Mycobacterium bovis* is its primary causative agent [1,3,4,5]. Other members of *Mycobacterium tuberculosis* complex (MTBC), such as *M. tuberculosis*, have also been isolated from naturally infected cattle [1,3,4,6,7]. Regardless of the applied control strategies, bTB is still prevalent in many developing countries, such as Burkina Faso and Ethiopia [8,9,10]. Yet knowledge is still scarce so far on its molecular epidemiology in Sub-Saharan African endemic areas, calling for implementation of surveillance approaches in order to better understand its local dynamics.

bTB has considerable economic and public health importance [11]. It causes economic losses of up to USD 3 billion each year worldwide [11,12]. In addition, bTB plays a significant role in human TB with probably up to 10% of human TB cases being ascribed to zoonotic transmission globally [13]. In 2018, the WHO reported that a total of 143,000 incident TB cases in humans were due to *M. bovis* worldwide [14]. Humans usually become infected by direct contact with infected animals or by consumption of contaminated raw meat and non-pasteurized milk [1,3,4,5]. Such infections of humans with *M. bovis* have repeatedly been reported in Africa (e.g., in Ethiopia and Kenya) [6,15].

Apart from the global point of view, bTB is also endemic in many parts of Africa [10]. It has been diagnosed in cattle and other animals with varying prevalence rates [10]. The vast majority of the African human and cattle populations live in areas where bTB is widely uncontrolled and its distribution is not monitored by surveillance approaches [4,5,16]. Therefore, the risk of *M. bovis* transmission to humans could be considerable in these areas.

In resource-limited Burkina Faso, in particular, bTB is considered as one of the important animal diseases [10]. Since about 80 years ago, a small number of studies have investigated bTB, mostly in Bobo-Dioulasso and Ouagadougou, as recently reviewed by Dibaba et al. [10]. These studies reported a prevalence of 0.13–27.7% of bTB in cattle applying tuberculin skin testing (TST) or routine postmortem inspections in slaughterhouses [10]. More in-depth information on the molecular characteristics of the causal agents of bTB in Burkina Faso are, however, scarce. Therefore, the aim of our study was to determine the prevalence of bTB suggestive lesions in slaughtered cattle carcasses as well as to characterize the *Mycobacterium* species isolated from these bTB suggestive lesions in two regions of Burkina Faso.

## 2. Materials and Methods

### 2.1. Study Setting and Design

This cross-sectional study was conducted in the period between August and September 2013, in two selected regions in Burkina Faso, namely Oudalan (Sahel region) and Ouagadougou (Central region) (Appendix A Figure A1).

Oudalan province is one of the four provinces of the Sahel region and is located in the northern part of the country and is divided into seven administrative units. It shares international borders with Niger and Mali and national borders with Séno from the south and Soum from the west (Figure A1). The province has many water sources, and animals from Niger and Mali cross the international borders into Oudalan, Burkina Faso, in search of water and grass and intermingle with local animals during dry seasons. This intermingling offers an ideal condition for transboundary animal and human disease transmission among animals and humans from the three neighboring countries. The Central region is the most populous and urbanized region of Burkina Faso and has only one province “Kadiogo“, which is surrounded by five provinces, including Bazéga, Boulkiemdé, Kourwéogo, Ourbritenga, and Ganzourgou from the south, the west, the north, and the east (Figure A1).

The country has a cattle population of 8 million heads [17]. The Sahel region was included in the study as it harbors 25% of the country’s cattle population (i.e., the largest cattle population in comparison with other regions). In addition, the region has the 3rd highest human TB burden in the country. The extensive traditional animal production system prevails in the assessed regions of the country [17]. Cattle owners and breeders live in close contact with their animals and often consume raw milk. This habit poses a high risk of contracting bTB when consuming products from an infected animal. On the other hand, the Central region was included in the study because of the high number of slaughtered animals, as it has the largest slaughterhouses in the country.

In total, seven (i.e., one in each administrative unit) and one (i.e., the largest animal processing plant in Burkina Faso) slaughterhouses in Oudalan and Ouagadougou were investigated, respectively.

### 2.2. Sample Size Calculation and Sampling Strategy

Using the following standard formula: *n* = (1.96)^2^ × P_exp_ × (1 − P_exp_)/d^2^ (whereby, *n* is the required sample size, (1.96)^2^ is the statistic corresponding to level of confidence, P_exp_ is the expected bTB prevalence of up to 27.7% according to Dibaba et al. [10], and d is desired absolute precision [±5%]), the calculated *n* was 308 animals from each of the two investigated regions [18]. This number was inflated 3-fold to account for the effect of randomness and representativeness [18]. Thus, total *n* was 1152 animals from each slaughterhouse and 2304 animals from both. Nevertheless, 2165 carcasses were investigated during the study.

Study animals were selected by systematic random sampling during antemortem inspection according to Thrusfield [18]. Selected animals were identified using a permanent marker/tag and were kept in a separate pen until slaughtering time. The sampling interval was obtained by dividing the total number of slaughtered animals on that day by the estimated daily sample size [18].

### 2.3. Antemortem and Postmortem Inspection

Antemortem and postmortem inspections were performed according to the standard procedure described elsewhere [19,20]. Briefly, all animals were collectively and individually examined. During antemortem inspection, the examination focused on observing the animals’ general behavior, nutritional status (i.e., body condition), cleanliness of the natural orifices for presence of nasal discharge due to respiratory diseases or for diarrhea and exudates, and for other clearly visible signs of disease and abnormalities. Postmortem inspection was by visual observation, physical palpation, and in situ slicing of lymph nodes, organs, and muscles of the head, the thoracic and the abdomen cavities, as well as the whole carcass and joints. Carcasses of the animals were considered as bTB suspected cases whenever gross lesions suggestive of bTB were detected.

### 2.4. Sample Collection and Transportation

Samples were collected from lymph nodes, tissues and organs, and muscles of the carcasses with bTB suggestive lesions by using sterile surgical scalpels and scissors [19]. Around 70 g was taken from each bTB suggestive lesion and placed in a sterile container, which was properly labeled and placed in an ice box for transportation to the National Laboratory of Veterinary Services in Ouagadougou, Burkina Faso, where samples were frozen at −4 °C until transportation to the National Reference Center for Mycobacteria in Borstel, Germany. 

### 2.5. Laboratory Procedures

Samples were subjected to various laboratory techniques, including decontamination, smear microscopy, mycobacterial culture, and molecular typing at the National Reference Center for Mycobacteria in Borstel, Germany.

#### 2.5.1. Decontamination of the Samples

bTB suggestive lesions collected from suspected tissue were homogenized using the ULTRA-TURRAX^®^ tube driver or homogenizing and dispersing device (IKA^®^-Werke GmbH & Co. KG, Staufen, Germany). Samples were prepared for culture by decontamination with sodium hydroxide and the mucolytic agent *N*-acetyl-L-cysteine (NALC-NaOH) [21]. For this, an equal volume of NALC-NaOH was added to the homogenized sample. Subsequently, the mixture was incubated for 20 min on a shaker, then 10 to 20 mL sterile phosphate buffer and 1 to 2 drops of 5.0% Tween 80 were added and centrifuged for 20 to 30 min (3500× *g*). Finally, the supernatant was poured off and the pellet was suspended in 1.0 mL sterile phosphate buffer.

#### 2.5.2. Smear Microscopy

Smears were stained with Kinyoun stain as described by GLI [21]. A volume of 20 µL of the resuspended granulomatous tissue and sputum pellets were used for smear microscopy. It was dropped on a clean slide, air dried, heated for fixation, and stained using an automated staining system (ZN Aerospray^®^ TB Slide Stainer/Cytocentrifuge, Wescor, Logan, UT, USA), and read using light microscopy (oil immersion lens, 1000× magnification). Results were recorded as smear positive or smear negative. All smear positive samples were graded as +/−, 1+, 2+, 3+.

#### 2.5.3. Mycobacterial Culture

The decontaminated and resuspended bTB suggestive lesions were aseptically inoculated into mycobacterial growth indicator vials (MGIT; Becton-Dickinson, Franklin Lakes, NJ, USA) primed with growth supplement and antibiotics (PANTA™; Becton-Dickinson) [21,22]. The MGIT tubes were incubated in the BD MGIT 960 instrument for a maximum of 42 days. In addition, culture on two Löwenstein-Jensen, two Stonebrink and one Middlebrook 7H10 slopes with antibiotic supplement (Enclit, Oelzschau, Leipzig, Germany) was performed at 37 °C for a maximum of 56 days.

#### 2.5.4. Line Probe Assay

The commercially available line probe assays (LPA) GenoType *Mycobacterium* CM (common mycobacteria) and GenoType *Mycobacterium* MTBC (*Mycobactrium tuberculosis* Complex) were conducted according to the instructions of the manufacturer (Hain Lifescience GmbH, Nehren, Germany). First, a volume of 1 mL of the liquid culture medium was centrifuged (10,000× *g* for 15 min at room temperature), the supernatant was discarded, and the pellet was suspended in 300 to 500 µL of distilled water. The suspended pellet was boiled for 20 min, then sonicated for 15 min for DNA extraction.

The LPAs were performed by using 35 µL of a primer-nucleotide mixture (provided with the kit), amplification buffer containing 2.5 mM MgCl_2_, 1.25 U of hotstart *Taq* polymerase (QIAGEN, Hilden, Germany), and 5 µL of the heat-inactivated suspension giving a final volume of 50 µL for amplification. The amplification protocol consisted of 15 min of denaturing at 95 °C, followed by 10 cycles comprising 30 s at 95 °C and 120 s at 58 °C, an additional 20 cycles comprising 25 s at 95 °C, 40 s at 53 °C, and 40 s at 70 °C, and a final extension at 70 °C for 8 min. Hybridization and detection were performed in an automated washing and shaking device (Profiblot; Tekan, Maennedorf, Switzerland). The program was started after mixing 20 µL of the amplification products with 20 µL of denaturing reagent (provided with the kit) for 5 min in separate troughs of a plastic well. Automatically, 1 mL of prewarmed hybridization buffer was added, followed by a stop to put the membrane strips into each trough. The hybridization procedure was performed at 45 °C for 0.5 h, followed by two washing steps. For colorimetric detection of hybridized amplicons, streptavidin conjugated with alkaline phosphatase and substrate buffer was added. After the final washing, strips were air dried and fixed on paper.

Each GenoType CM strip contains 17 probes, including amplification and hybridization controls to verify the test procedure. With the CM assay, 15 patterns can be obtained from 23 species (10 individually and 13 in combination). The GenoType MTBC guarantees reliable differentiation between the strains of *M. tuberculosis/M. canettii*, *M. bovis*, *M. bovis* Bacillus Calmette-Guérin (BCG), *M. caprae*, *M. africanum*, and *M. microti*.

#### 2.5.5. Sanger Sequencing

For further identification of the non-tuberculous mycobacteria (NTMs), sequencing of the 16S was carried out. Amplification of the DNA using 264 primers A and B was performed. The complete PCR products were sequenced on an automated DNA sequencer (ABI 377; Applied Biosystems, Waltham, MA, USA) by cycle sequencing using the Big Dye RR Terminator cycle sequencing kit (Applied Biosystems). The resulting sequences were aligned and compared to the sequences of the International Nucleotide Sequence Database Collaboration.

#### 2.5.6. Spoligotyping

The spoligotyping was performed according to the spoligotype kit supplier’s instructions (Ocimum Biosolutions Company, Ijsselstein, The Netherlands). The DR region was amplified by PCR using oligonucleotide primers (DRa: 5′ GGTTTTGGG CTGACGAC 3′ and DRb: 5′ CCGAGAGGGGACGGAAAC 3′) derived from the mycobacterial DR locus sequence. The DRa primer is biotinylated at the 5′-end. A total volume of 25 μL of the following reaction mixture was used for the PCR: 12.5 μL of HotStarTaq Master Mix (Qiagen: this solution provides a final concentration of 1.5 mM MgCl2 and 200 μM of each deoxynucleotides triphosphates), 2 μL of each primer (20 pmol each), 5 μL suspension of heat-inactivated cells (approximately 10 to 50 ng), and 3.5 μL distilled water. The mixture was heated for 15 min at 96 °C and subjected to 30 cycles of 1 min at 96 °C, 1 min at 55 °C, and 30 s at 72 °C. The amplified product was hybridized to a set of 43 immobilized oligonucleotides, each corresponding to one of the unique spacer DNA sequences within the DR locus. After hybridization, the membrane was washed twice for 10 min in 2× SSPE (1× SSPE is 0.18 M NaCl, 10 mM NaH2PO4, and 1 mM EDTA [pH 7.7])/0.5% sodium dodecyl sulphate at 60 °C and then incubated in 1:4000 diluted streptavidin peroxidase (Boehringer, Ingelheim am Rhein, Germany) for 45 to 60 min at 42 °C. The hybridized DNA was detected by an enhanced chemiluminescence method (Amersham) and by exposure to an X-ray film (Hyperfilm ECL, Amersham, The Netherlands).

#### 2.5.7. 24-Loci MIRU-VNTR Typing

The 24-loci MIRU-VNTR typing was conducted according to a standardized protocol [23]. Briefly, MIRU-VNTR alleles were amplified using Quadruplex PCR Kit (Genoscreen, Lille, France). Fragment analysis using the GeneScan™ 1200 LIZ dye as a size standard (Life Technologies, Darmstadt, Germany) was carried out on a capillary sequencer 3130xL and 3500xL for the genetic analyzer. The GeneMapper software version 3.7 (Life Technologies, Darmstadt, Germany) was used to determine the copy number of MIRU-VNTR alleles.

### 2.6. Statistical Analysis

Due to the low number of positive samples, data were solely descriptively presented.

## 3. Results

### 3.1. Study Population

The study population consisted of all cattle slaughtered at the Ouagadougou and Oudalan slaughterhouses/slabs during the study period. In total, 20,374 cattle were slaughtered; of that, 80.6% of these cattle were slaughtered at the slaughterhouses of the Central region and the rest (19.4%) were slaughtered at the slaughterhouses/slabs of the Sahel region. The male to female ratio was 1:2, and only animals older than two years were slaughtered. The animals were mostly of indigenous breeds, e.g., Lobi, Doayo, and N’Dama breed, as well as a small proportion of exogenous breeds from neighboring countries, for example, Azawak, Goudali, Djelli, and Bororo breeds from Niger and Mali.

### 3.2. Prevalence and Anatomical Site of Tuberculous Lesions

A total of 2165 cattle carcasses were investigated in this study, of which 58 carcasses had bTB suggestive lesions (i.e., tuberculous lesions) giving an overall prevalence of 2.7% (58/2165) with a 95% confidence interval (CI) of 2.7–3.5%.

The majority of the bTB positive cattle carcasses (77.6%, *n* = 45) were found at the Ouagadougou slaughterhouse and the remaining (22.4%, *n* = 13) were observed at Oudalan slaughterhouses and slabs. Moreover, the 45 bTB positive carcasses that were diagnosed at the Ouagadougou slaughterhouse originated from different provinces as follows: 17 cattle from Kadiogo, 8 from Soum, 7 from Gourma, 3 from each Sanmentenga and Kouritenga, and 1 from each of Bcle du Mouhoun, Houet, Sissili, and Ziro (Figure 1). The origin of the animals of the remaining three carcasses was unknown.

Stratifying the bTB positive carcasses by age and sex of the slaughtered cattle showed that 71.1% (*n* = 32; 95 CI 56.6–82.3%) were carcasses of over 6-year-old cattle, 24.4% (*n* = 11; 95 CI 14.2–38.7%) were carcasses of cattle between 2 and 6 years old, and only 2.2% (*n* = 1; 95 CI 0.39–11.6%) were carcasses of 2 years old cattle. The bTB positive carcasses of bulls and cows accounted for 60.3% and 39.7%, respectively.

A table detailing the characteristics of the animals with suspicious lesions is provided below (Table 1).

Regarding the anatomical localization of the bTB suggestive lesions, the majority of positive carcasses/cattle (96.6%, *n* = 56) had localized lesions confined to one or a few organs with the majority being detected in the visceral lymph nodes (56.9%, *n* = 33) and the lungs and thoracic lymph nodes (20.7%, *n* = 12) (Table 2). Only 3.4% (2/58) of the carcasses were observed with generalized lesions (i.e., miliary bTB, which is defined as the widespread dissemination of lesions in many organs or all over the body of the animal/carcass).

The detected bTB tubercle or granulomatous lesions had the typical appearance of caseous necrosis (e.g., seen in the lungs and liver) (Figure 2). Other lesions were whitish and yellowish of various sizes congregating altogether or scattered and enclosed in light congested grey fibrous tissue.

### 3.3. Isolation and Species Identification

Culturing of the samples collected from bTB positive animals revealed growth of *Mycobacterium* species from 77.6% (45/58) specimens. The remaining 13 cultures were either extensively contaminated showing growth of other bacteria and yeasts (1.7%, 1/58) or negative (i.e., no mycobacterial growth could be observed in any of the five cultures) (22.4%, 13/58).

Molecular testing of the grown mycobacteria by using LPAs GenoType *Mycobacterium* CM, GenoType *Mycobacterium* MTBC, and 16S sequencing revealed pure cultures of *M. bovis* (44.4%, *n* = 20), *M. fortuitum* (8.9%, *n* = 4), *M. elephantis* (6.7%, *n* = 3), *M. brumae* (4.4%, *n* = 2), *M. avium* (2.2%, *n* = 1), *M. asiaticum* (2.2%, *n* = 1), *M. terrae complex* (2.2%, *n* = 1), and unknown NTMs (11.1%, *n* = 5) (Table 3). However, mixed cultures with more than one *Mycobacterium* species identified were also observed (e.g., three mixed cultures of *M. bovis* and *M. fortuitum* and three mixed cultures of *M. bovis* and *M. elephantis*) (Table 3).

### 3.4. Molecular Typing of the MTBC Strains

Spoligotyping and 24-loci MIRU-VNTR typing results were available for all of the 26 *M. bovis* strains, including the mixed cultures of *M. bovis* and *M. fortuitum* as well as *M. bovis* and *M. elephantis*. The combined results of the two typing methods classified the strains as *M. bovis* (Figure 3). The MTBC strain that was mixed with *M. kubicae* was not characterized.

As Table 4 shows, the detected *M. bovis* strains were associated with 14 different MLVA 15-9 codes ranging in frequency from 1 to 10. However, the MLVA 15-9 code of one strain was not complete due to a missing MIRU-VNTR copy and/or a spacer. *M. bovis* strains with identical MIRU-VNTR and spoligotyping profiles were assigned to molecular clusters as surrogate markers for putative transmission networks. Each cluster was assigned to a unique MLVA 15-9 code using the MIRU-VNTR*plus* online database [24]. Overall, 13 strains were assigned to 6 clusters that comprised 2–42 strains or patients resulting in a clustering rate of 50.0%. The *M. bovis* strains with 7-7, MLVA 15-9 codes were most prevalent and accounted for 50.0% (*n* = 13) of all *M. bovis* strains (Table 4).

Combined characterization of the *M. bovis* strains by using spoligotyping and 24-loci MIRU-VNTR typing showed that five carcasses of animals were infected with *M. bovis* clones that had different MLVA 15-9 codes pointing to mixed *M. bovis* infections as follows: 7-7 and 14119-7, 7-7 and 14117-7, 319-7 and 7-1140, as well as 7-7 and 14117-7.

## 4. Discussion

This study was conducted to estimate the prevalence of mycobacterial infections in cattle in Burkina Faso and to investigate the molecular epidemiology of the isolated strains. Overall, a prevalence of 2.7% was observed, with considerable geographic variation ranging from 2.6% in Ouagadougou to 16.6% in Oudalan. More than half of the 58 identified animals with bTB suggestive lesions were infected with MTBC strains. The observed bTB lesions were restricted to a few organs in the vast majority of cases, calling for thorough veterinary examinations of cattle carcasses being processed for human consumption.

The Sahel region of Africa (including Burkina Faso) offers ideal conditions for livestock breeding because of its eco-climatic and bio-geographic characteristics [25]. However, the development and exploitation of livestock herds and flocks for the benefit of the people in countries of the Sahel region is hindered by many challenges, e.g., drought and many endemic diseases such as bTB [25,26]. In this study, 2.7% of the investigated cattle carcasses showed bTB suggestive lesions. Former studies reported a prevalence of bTB suggestive lesions of up to 6.8% in Burkina Faso [9,27,28]. In livestock species other than cattle, 1.07% of the small ruminants and 0.7% of the pigs investigated at Bobo-Dioulasso slaughterhouse had bTB suggestive lesions [25]. In countries in Africa, e.g., Morocco, Algeria, Sudan, Uganda, and Ethiopia, the prevalence of visible gross bTB suggestive lesions was found to be as high as 5.2% [19,29,30,31,32,33]. By using other bTB-specific diagnostic methods, our study detected a high regional prevalence in Oudalan well in line with the results of a previous regional assessment [34]. In 2003, 18.6% infected cattle were recorded in the neighboring country Mali using a tuberculin-testing based diagnostic approach [34]. In a similar tuberculin testing-based study in Ougadougou in 2004, even a prevalence of 27.7% was recorded [35], which is considerably higher than in the present study, confirming that immune-conversion does not necessarily indicate clinically apparent infection. Microscopic testing from those lesions showed visible mycobacteria in 54% of the instances. Mycobacterial culture was attempted with a predominance of *M. bovis* similar to the present study. A slightly lower positivity rate of 25.9% mycobacterial detections from tuberculous lesions in tissue of slaughtered cattle as assessed by microscopy and culture was reported from Ethiopia in 2010 [36]. Altogether, the results of the present assessment fit well in the previously described literature results and suggest that bTB was an ongoing menace in Burkina Faso at the time of the present assessment.

The anatomical localization of bTB lesions can vary [37,38,39]. Depending on the severity of the disease, bTB lesions can be localized and confined to one or a few organs, such as the lungs and/or the liver and their lymph nodes [37,38,39]. However, in severe cases, the bTB lesions can be generalized, defining a variant of the disease known as miliary bTB. The spread of the *M. bovis* leading to generalized bTB lesions occurs via the animals’ blood stream [37,38,39]. In the present study, most of the infected cattle had confined bTB lesions indicating that bTB is less severe in Burkina Faso.

In line with previous studies [34,35], a higher age of more than two years of the animals was identified as a risk factor for bTB in the present study. This finding is not quite surprising, because longer exposition time is likely associated with increased probability of getting infected. Interestingly, a non-significant tendency for a higher prevalence in male than in female animals was not only observed in the present study but also confirms previous results from other investigators [34,35]. The recorded predominance of *M. bovis* among the mycobacteria isolated is also well in line with other regional studies [35,40]. While *M. avium* comprised most of the detected NTMs in cattle in Tchad [40], this study found that *M. fortuitum* and unknown NTMs besides *M. elephantis* represent the majority of the NTMS. Moreover, *M. brumae*, which is an environmental, non-pathogenic, rapidly growing, non-chromogenic *Mycobacterium* species with immunotherapeutic capacities, was also found in this study [41].

Based on the global spoligotype patterns’ diversity, provided as conserved on the international database on spolygotyping (http://www.mbovis.org, last accessed on 12 May 2022), a small number of different *M. bovis* spoligotypes but also a considerable number of non-typable strains were identified. Among the identified spoligotypes, SB044 was the most frequently observed with a proportion of 31.4%. This spoligotype had previously been reported from Mali, Nigeria, Cameroon, and Chad [42,43,44], confirming the widespread distribution of SB044 in West and Central Africa. Spoligotype B0300, which was identified only once in the present study, was detected in Mali before [42,43,44] and SB0328, which was similarly rare in the present assessment, in Chad and Nigeria [42,43,44]. Migration and transhumance of animals, whereby animals are mixed and exchange their microbiomes with each other, are likely to contribute to the geographical distribution of such repeatedly identified spoligotypes.

The study has a number of limitations. Only visible lesions were included in the microbiological assessments. So, it is likely that infections with macroscopically undetectable lesions will have been missed, resulting in an underestimation of immunologically relevant contacts to mycobacteria. Furthermore, no human samples were included in the study. In a recent study from Burkina Faso, clonally related *M. bovis* strains of the clonal complexes African 1 (Af1) and African 5 (Af5) were identified both in cattle and in human patients [9], suggesting relevant transmission links. However, due to the lack of human samples, zoonotic transmission could not be investigated herein. Another limitation of the study is the fact that modern biosecurity scores [45], recently included in a European study, were not applied for the assessment. The application of such standardized scores generally facilitates the comparison of studies on rare diseases such as bovine tuberculosis, which still occurs in a one-digit percentage range even in endemicity areas in Europe [46]. In addition, the fact that sampling was conducted nine years ago limits the interpretation of the data. So, it cannot be excluded that epidemiological shifts have occurred in the meantime.

## 5. Conclusions

This study confirmed the prevalence of mycobacteria in cattle in Burkina Faso with high regional variance. *M. bovis* is the predominant causative agent with the spoligotype SB044 being most frequently isolated. Future surveillance should cover larger areas, should occur continuously, and also include human sample materials to provide more reliable information on the local epidemiology of bTB. In spite of the abovementioned limitations, the study indicated an ongoing need for preventive action in order to avoid the marketing and consumption of infected meat in Burkina Faso. Further, the transborder-migration of mycobacterial clones, as indicated by spoligotyping, calls for continuous surveillance and typing approaches in order to identify and inhibit the source of mycobacterial spread in Western Africa. Nevertheless, spoligotyping has laid a foundation for future investigations to clarify the role of transboundary animal trade, e.g., to Ghana, and wildlife contacts in the spread of animal mycobacterioses.

## Figures and Tables

**Figure 1 microorganisms-10-01378-f001:**
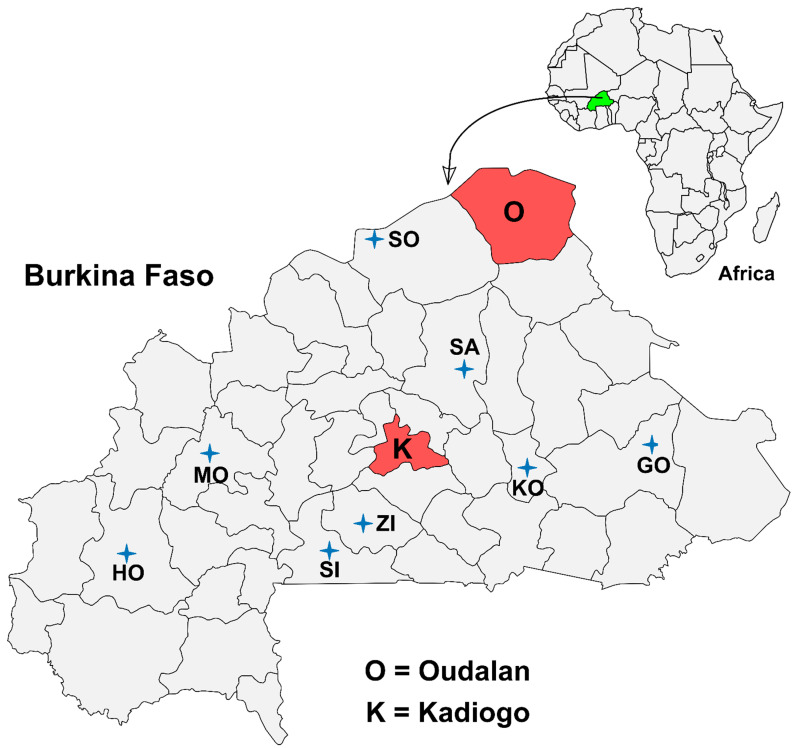
Map of Burkina Faso showing the study area and the origin of cattle with bTB suggestive lesions (i.e., bTB positive) that were processed for meat for human consumption. The sampling sites “O” (Oudalan) and “K” (Kadiogo) provinces are highlighted in reddish color. bTB = bovine tuberculosis, SO = Soum, SA = Sanmentenga, GO = Gourma, KO= Kouritenga, ZI = Ziro, SI = Sissili, HO = Houet, and MO= Mouhoun. The stars indicate the positions of the towns.

**Figure 2 microorganisms-10-01378-f002:**
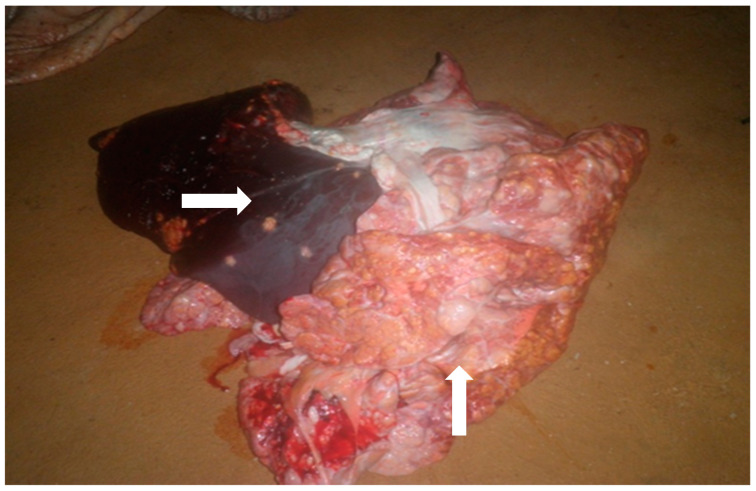
Gross appearance of granulomatous bTB suggestive lesions in the liver and lungs of a slaughtered cattle in Burkina Faso (indicated by white arrows). bTB = bovine tuberculosis.

**Figure 3 microorganisms-10-01378-f003:**
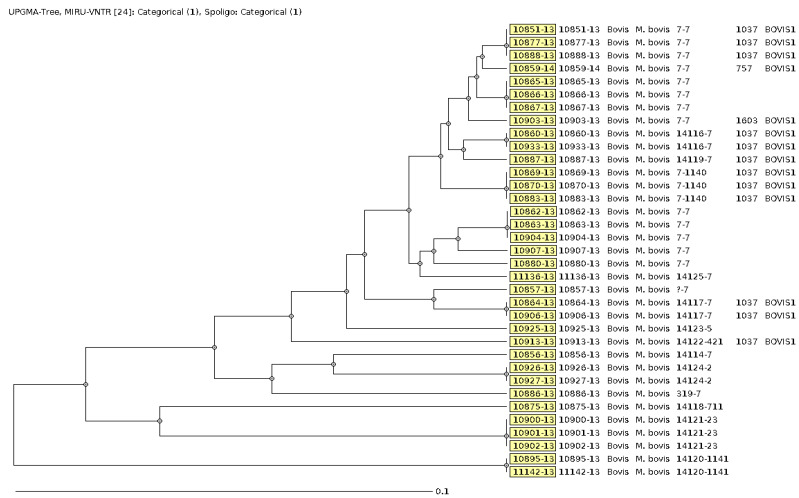
Genetic diversity (i.e., phylogenetic relationship) among *M. bovis* strains isolated and characterized. Unweighted pair group method with arithmetic mean (UPGMA) tree calculated using 35 strains.

**Table 1 microorganisms-10-01378-t001:** Characteristics of the animals with suspicious lesions.

	All, *n* = 58		
Characteristics	Number	Percentage %	IC 95%
Lesion’s localization			
generalized to the entire carcass	2	3.5	[0.4–12.1%]
Localized in one or a few organs	56	96.5	[87.9–99.6%]
Age			
Young (less than 2 years)	1	1.8	[0.0–9.6%]
Young adult (between 2 and 6 years	15	26.8	[15.8–40.3%]
Adult (over 6 year)	40	71.4	[57.8–82.7%]
Sex			
Male	33	58.9	[45.0–71.9%]
Female	23	41.1	[28.1–55.0%]
Slaughterhouse			
Ougadougou	45	77.6%	n.a.
Oudalan	13	22.4%	n.a.

n.a. = not applicable.

**Table 2 microorganisms-10-01378-t002:** Frequency of body organ(s) with bTB lesions.

Organ(s) with bTB Lesion	Number	%
Liver	6	10.3
Lungs	2	3.4
Lungs and thoracic LNs	12	20.7
Lungs, liver, and thoracic and hepatic LNs	2	3.4
Visceral LNs	33	56.9
Liver and hepatic LNs	1	1.7
Whole body (miliary bTB lesions)	2	3.4
Total	58	100

bTB = bovine tuberculosis, LNs = lymph nodes.

**Table 3 microorganisms-10-01378-t003:** Frequency of detected *Mycobacterium* species.

*Mycobacterium* Species	Number	%
*M. bovis*	20	44.4
*M. bovis* and *M. fortuitum*	3	6.7
*M. bovis* and *M. elephantis*	3	6.7
*M. fortuitum*	4	8.9
*M. elephantis*	3	6.7
*M. brumae*	2	4.4
*M. avium*	1	2.2
*M. asiaticum*	1	2.2
*M. terrae complex*	1	2.2
MTBC and *M. Kubicae*	1	2.2
Unknown NTM and *M. novocastrense*	1	2.2
Unknown NTM	5	11.1
Total	45	100

MTBC = *Mycobacterium tuberculosis* complex, NTM = non-tuberculous mycobacteria.

**Table 4 microorganisms-10-01378-t004:** MLVA 15-9 codes of the *M. bovis* strains.

MLVA 15-9 Code	Number
7-7	10
319-7	1
14114-7	1
14116-7	2
14117-7	2
14119-7	1
14121-23	1
14123-5	1
14124-2	2
14125-7	1
14122-421	1
14118-711	1
14120-1141	1
?-7 *	1

* = Missing MIRU-VNTR copy and/or a spacer.

## Data Availability

Not applicable.

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
