# Peer review of "Prevalence and Molecular Characterization of Mycobacterium bovis in Slaughtered Cattle Carcasses in Burkina Faso; West Africa"

_microorganisms, 2022, doi:10.3390/microorganisms10071378_

Round 1

Reviewer 1 Report

The aim of our study was to determine the prevalence of bTB suggestive lesions in slaughtered cattle carcasses as well as to characterize the Mycobacterium species isolated from these bTB suggestive lesions in two regions of Burkina Faso. The study is well designed

- abstract is long, please follow the journal guidelines.

- Please revise the manuscript for English.

- No need for figure 1. Please delete.

- Provide the Ethical Statement with the protocol; number included.

- Add a table in material and methods to distribute your samples according to sex, age, breed, sampling location etc.

- Line 259: Statistical analysis.

- Combine the information in both maps in one map and delete the other map.

- Why do you keep writing this (Prevalence and molecular characterization of Mycobacterium bovis in slaughtered cattle carcasses in Burkina Faso) many times in the text and tables title?

- no mycobacterial growth could be observed in any of the five cultures: was the animal source of these samples showed lesions?

- Please highlight the statistical analysis in your results.

Author Response

Reviewer one

The aim of our study was to determine the prevalence of bTB suggestive lesions in slaughtered cattle carcasses as well as to characterize the Mycobacterium species isolated from these bTB suggestive lesions in two regions of Burkina Faso. The study is well designed

- abstract is long, please follow the journal guidelines.

We thank the reviewer for this hint. In response to the comments by reviewer 2, however, we even had to amend the abstract by another sentence. We respectfully ask the editor to accommodate this. Instead, we have removed two sentences on methodical details.

- Please revise the manuscript for English.

We thank the reviewer for this hint. The manuscript was revised.

- No need for figure 1. Please delete.

We agree that there is no feed for this figure in the main text. As, however, we believe that it nicely illustrates the geographic location of the study area, please allow us to shift it to the appendix A as a help for readers who are not well familiar with the local geography of Burkina Faso.

- Provide the Ethical Statement with the protocol; number included.

The samples were collected at a slaughter house, from animals that were already dead. So, no Ethical statement was necessary. However, the first author of the study, Estelle Kanyala, who collected the samples, applied for an official permission to collect the samples, before collecting the samples. This official permission will be uploaded to the journal.

- Add a table in material and methods to distribute your samples according to sex, age, breed, sampling location etc.

As requested, with have added a table with available respective information as a new table 1. We have decided to present this table in the results chapter “3.2. Prevalence and anatomical site of tuberculous lesions“, because we feel that it better fits there than in the methods section. We respectfully ask you to accommodate this decision.

- Line 259: Statistical analysis.

The Heading “data analysis” was changed to “Statistical analysis”

- Combine the information in both maps in one map and delete the other map.

In line with your suggestion above, we have solved the issue by transferring the previous figure 1 to the appendix paragraph. In this way, the contents of both maps can be preserved without unnecessarily making the paper exceedingly long.

- Why do you keep writing this (Prevalence and molecular characterization of Mycobacterium bovis in slaughtered cattle carcasses in Burkina Faso) many times in the text and tables title?

We thank the reviewer for this hint. This was changed.

- no mycobacterial growth could be observed in any of the five cultures: was the animal source of these samples showed lesions?

Yes. However, the samples were collected according tie the aspect in visual inspection. The lesions may therefore have another cause than mycobacteria.

- Please highlight the statistical analysis in your results.

Due to the low number of positive samples, we decided to provide a descriptive presentation of the results only, although more elaborated statistics had been initially planned. We have rephrased the “Statistical analysis” sub-heading of the methods chapter accordingly.

Reviewer 2 Report

The title is nice and adequately reflect the study, however, add “West Africa” at the end.

The abstract adequately summarize methodology, results, and significance of the study. However authors should add statistical analysis applied in the study.

The introduction section is well structured and it falls within the topic of the study. Punctuation should be improved. Also English language should be ameliorated.

I suggest to add some information relating to the importance of  biosecurity score on the prevalence of infectious diseases on dairy farms (please, read and add the citation: Perillo L., et al., Journal of Veterinary Research (Poland), 2022, 66 (1), 103 – 107).

Also, please add latest reference on the field (Abbate J.M., et al., Animals, 2020, 10(9): 1473).

The material and methods section is well written and Authors meticulously describe the methods applied in the study. However, I have some questions for the Authors:

Did Authors select inclusion or exclusion criteria for the selection of the herd?

Regarding statistical analysis, Did Authors assess the normal distribution of data by Normality test?

Results as well as discussion section are well presented, and Authors well discuss the findings obtained in their study reporting appropriate bibliographic information.

The conclusion section is well written, however, the section is too short. Authors should better summarize the main findings and emphasize the significance of the study.

The tables as well as Figures are generally good they well represent the study. I suggest to delete the Figure 2 as it lacks of significance.

Authors should check and standardize the references in the list according to journal guidelines.

Author Response

Reviewer 2

The title is nice and adequately reflect the study, however, add “West Africa” at the end.

West Africa was added.

The abstract adequately summarize methodology, results, and significance of the study. However authors should add statistical analysis applied in the study.

We thank the reviewer for that hint. It was added that data were just descriptively assessed due to the low number of positive samples.

The introduction section is well structured and it falls within the topic of the study. Punctuation should be improved. Also English language should be ameliorated.

We thank the reviewer for that hint. The manuscript was thoroughly proofread once again prior to resubmission.

I suggest to add some information relating to the importance of  biosecurity score on the prevalence of infectious diseases on dairy farms (please, read and add the citation: Perillo L., et al., Journal of Veterinary Research (Poland), 2022, 66 (1), 103 – 107).

Also, please add latest reference on the field (Abbate J.M., et al., Animals, 2020, 10(9): 1473).

As requested, we have added your points in the limitations paragraph of the discussion.

Please adapt the quotation to the style of MDPI Microorganisms and add the manuscripts to the list.

The stylistic adaptations have been performed as requested.

The material and methods section is well written and Authors meticulously describe the methods applied in the study.

We thank the reviewer for that comment.

However, I have some questions for the Authors:

Did Authors select inclusion or exclusion criteria for the selection of the herd?

All applied inclusion and exclusion criteria had already been mentioned. On other criteria applied.

Regarding statistical analysis, Did Authors assess the normal distribution of data by Normality test?

Due to the low number of positive results, the statistical assessment has been restricted to a descriptive presentation of the data. For this purpose, no normality testing is necessary.

Results as well as discussion section are well presented, and Authors well discuss the findings obtained in their study reporting appropriate bibliographic information.

We thank the reviewer for that comment.

The conclusion section is well written, however, the section is too short. Authors should better summarize the main findings and emphasize the significance of the study.

We discussed the significance in more detail in the conclusions paragraph. In detail, the need for preventive measure and surveillance as indicated by the study results was particularly emphasized. Nevertheless, spoligotyping has laid a foundation for future investigations to clarify the role of transboundary animal trade, e.g., to Ghana, and wildlife contacts in the spread of animal mycobacterioses.

The tables as well as Figures are generally good they well represent the study. I suggest to delete the Figure 2 as it lacks of significance.

We agree and have removed the superfluous figure 2.

Authors should check and standardize the references in the list according to journal guidelines.

The reference list was checked and corrected. In detail, the reference numbers were set in brackets and the use of bold and italics typing was adjusted, punctuation and commas were adapted.

Reviewer 3 Report

I enjoyed reading your article on Mycobacterium bovis prevalence and molecular characterization in Burkina Faso. The manuscript is well-written, and the authors did an excellent job analyzing the data. This reviewer, on the other hand, is concerned about the sampling done 9 years ago, which may not have provided information about the current status of bTB in Burkina Faso. There are also a few minor issues that must be addressed (please see attached file). 

Author Response

Reviewer 3

I enjoyed reading your article on Mycobacterium bovis prevalence and molecular characterization in Burkina Faso. The manuscript is well-written, and the authors did an excellent job analyzing the data. This reviewer, on the other hand, is concerned about the sampling done 9 years ago, which may not have provided information about the current status of bTB in Burkina Faso. There are also a few minor issues that must be addressed (please see attached file).

As requested, we have added the considerable age of the data as another limitation to the limitations paragraph at the end of the discussion. Also, we have corrected the wording issues as marked in the attached file.

Round 2

Reviewer 1 Report

The authors addressed all my comments,  however there still need for English revision